# X-ray Diffraction, Micro-Raman and X-ray Photoemission Spectroscopic Investigations for Hydrothermally Obtained Hybrid Compounds of Delafossite CuGaO$_2$ and Wurtzite ZnO

**Minuk Choi [1], Christoph Brabec [2] and Tomokatsu Hayakawa [1,3,\*]**

[1]   Field of Advanced Ceramics, Department of Life Science and Applied Chemistry, Nagoya Institute of Technology, Gokiso, Showa, Nagoya 466-8555, Japan

[2]   Department of Materials Science and Engineering, University of Erlangen-Nuremberg, Martensstraße 7, DE-91058 Erlangen, Germany

[3]   Frontier Research Institute of Materials Science (FRIMS), Nagoya Institute of Technology, Gokiso, Showa, Nagoya 466-8555, Japan

\*   Correspondence: hayatomo@nitech.ac.jp

**Abstract:** P-type delafossite CuGaO$_2$ is a wide-bandgap semiconductor for optoelectronic applications, and its lattice parameters are very similar to those of n-type semiconductor wurtzite ZnO. Accordingly, the investigation of crystalline heterostructures of CuGaO$_2$ and ZnO has attracted significant attention. In this study, interfacial CuGaO$_2$/ZnO hetero-compounds were examined through X-ray diffraction (XRD) analysis, confocal micro-Raman spectroscopy, and X-ray photo-electron spectroscopy (XPS). XRD and Raman analysis revealed that the hydrothermal deposition of ZnO on hexagonal platelet CuGaO$_2$ base crystals was successful, and the subsequent reduction process could induce a unique, unprecedented reaction between CuGaO$_2$ and ZnO, depending on the deposition parameters. XPS allowed the comparison of the binding energies (peak position and width) of the core level electrons of the constituents (Cu, Ga, Zn, and O) of the pristine CuGaO$_2$ single crystallites and interfacial CuGaO$_2$/ZnO hybrids. The presences of Cu$^{2+}$ ions and strained GaO$_6$ octahedra were the main characteristics of the CuGaO$_2$/ZnO hybrid interface. The XPS and modified Auger parameter analysis gave an insight into a specific polarization of the interface, promising for further development of CuGaO$_2$/ZnO hybrids.

**Keywords:** p-type semiconductor; hetero-interface; hydrothermal synthesis; hexagonal platelet hybrids; X-ray photoemission spectroscopy; Raman spectroscopy



## 1. Introduction

The development of semiconducting materials that can produce hydrogen [1], decompose organic pollutants [2], and improve power generation efficiency [3] toward environmental purification and energy management is a growing research field that has received significant attention from material chemists as well as device manufacturers [4,5]. Copper-based delafossite oxides with a composition of CuMO$_2$ (M = Al, Ga, Cr) have been studied as new p-type semiconductors because of their high hole mobility, nontoxicity, high abundance, environmental friendliness, and low cost [6].

Delafossite-type copper gallate (CuGaO$_2$) has a rhombohedral (3R) or hexagonal (2H) symmetry, with a layer of distorted GaO$_6$ octahedra which were sandwiched between linear O–Cu–O chains parallel to c-axis [7]. The valence band is composed of an electronic hybrid of the 3d orbital of Cu atoms and the 2p orbital of O atoms, which is delocalized by oxygen atoms and forms at a low energy level. In addition, Cu vacancies and interstitial O atoms can produce holes, resulting in p-type conductivity [8]. As a wide-band-gap semiconductor, CuGaO$_2$ possesses a direct band gap at high energy of 3.4–3.7 eV and a small absorption tail starting at 2 eV due to an indirect band gap [9], which enables

its applications in p-n junction devices, p-type dye-sensitized solar cells, and photocatalysts [10–12]. Ehara [13] successfully fabricated transparent delafossite-type $CuGaO_2$ thin films for dye-sensitized solar cells by a sol-gel method. The Ga source materials were dissolved in nitrate or acetylacetonate sols, and the films prepared with acetylacetonate had a higher transmittance than those prepared with nitrate. Xu et al. [14] reported the formation of ZnO nanowires with n-type semiconductor properties on the surface of p-type delafossite, $CuGaO_2$. The multihorned composites of hexagonal platelet $CuGaO_2$ in 3R structure and ZnO nanowires were successfully fabricated by a hydrothermal method. The luminescence from ZnO nanowires and electron-hole recombination at the p-n junction interface were observed in the composites.

Figure 1 shows the crystal structures of delafossite 3R $CuGaO_2$ and wurtzite ZnO; their lattice parameters ($a$, $b$, $c$, $\alpha$, $\beta$, and $\gamma$) can be compared. Despite their different crystal structures, as shown in Figure 1a,b, the lattice parameters of the rhombohedral $CuGaO_2$ structure ($R\text{-}3m$) [7] are very similar to those of the hexagonal ZnO ($P6_3mc$) [15], as shown in Figure 1c,d. Particularly, Cu (or O) in the $c$ plane [(006) plane] is stacked to form triangles with a side length of 2.9770(8) Å in delafossite $CuGaO_2$, whereas Zn (or O) forms triangles with a side length of 3.25010(1) Å, as indicated by the yellow dashed line. Hexagons with side lengths of 1.7188 Å in $CuGaO_2$ and 1.8764 Å in ZnO structures [red line in Figure 1c] suggest the possibility to form $CuGaO_2$/ZnO (CGO/ZnO) heterostructure. Figure 2 shows a prospected electronic structure of p-n heterostructure as a model of $CuGaO_2$(p-type)-ZnO(n-type) hybrids. It is well known that in a hybrid p-n junction with different types of semiconductors having opposite carrier transport characteristics, band bending of the electronic structure occurs at the interface between the semiconductors [16]. In an n-type semiconductor, as shown in Figure 2i, electrons excited in the conduction band enter the semiconductor because of the potential slope generated by band bending, and the holes in the valence band move to the adjacent semiconductor interface. In contrast, photogenerated holes in the p-type semiconductor remain in the interior, whereas electrons move toward the interface and arrive at the adjacent semiconductor (see Figure 2ii). The continuous excitation of electron-hole pairs is possible at the interface; consequently, electrons and holes experience effective charge separation by the steep slope of the band potential [17,18]. In this case, a redox reaction is expected on each surface (involving electrons and holes on the n- and p-type sides, respectively) when semiconductor heterostructures act as catalysts under light illumination above the band-gap energy (see Figure 2iii).

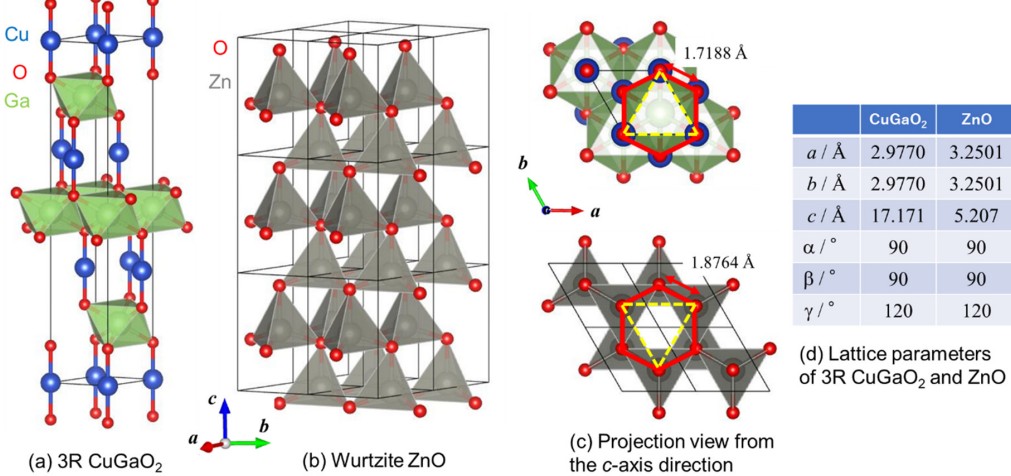

| | CuGaO₂ | ZnO |
|---|---|---|
| $a$ / Å | 2.9770 | 3.2501 |
| $b$ / Å | 2.9770 | 3.2501 |
| $c$ / Å | 17.171 | 5.207 |
| $\alpha$ / ° | 90 | 90 |
| $\beta$ / ° | 90 | 90 |
| $\gamma$ / ° | 120 | 120 |

(d) Lattice parameters of 3R CuGaO₂ and ZnO

(a) 3R CuGaO₂   (b) Wurtzite ZnO   (c) Projection view from the c-axis direction

**Figure 1.** Crystal structures of (**a**) rhombohedral (3R) $CuGaO_2$ and (**b**) wurtzite ZnO. (**c**) Projection views of 3R $CuGaO_2$ (**top**) and wurtzite ZnO (**bottom**). (**d**) Comparison of lattice parameters ($a$, $b$, $c$, $\alpha$, $\beta$, and $\gamma$) between 3R $CuGaO_2$ (PDF 01-082-8561) and ZnO (PDF 04-003-2106). Cu and Zn exhibit similar stacking in the $c$ plane, indicating the possibility to form a heterostructure between them via Cu–O–Zn bonding in the $c$-axis direction.

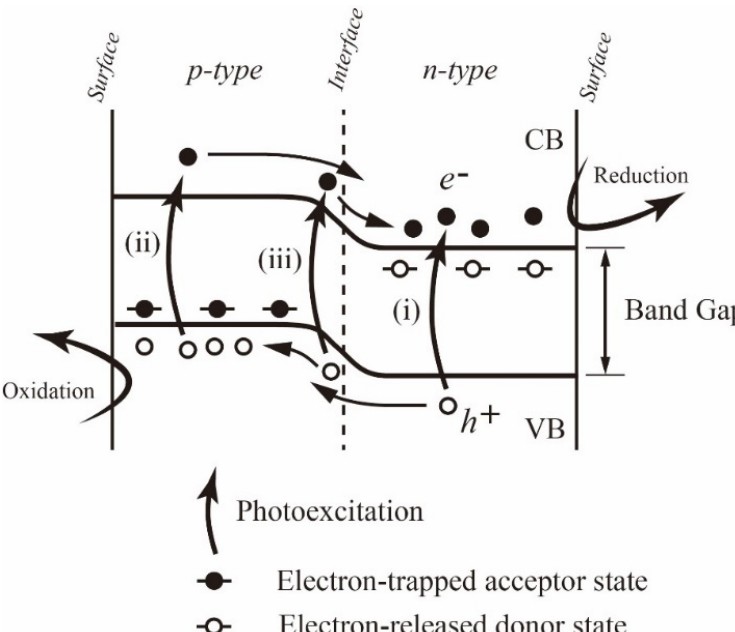

**Figure 2.** Schematic illustration of photocatalysis mechanism of a heterostructure using p- and n-type semiconductors under light illumination above band-gap energy. The potential slope at the interface between p- and n-type semiconductors can effectively separate generated electron-hole ($e^{-}$–$h^{+}$) pairs. VB: valence band, CB, conduction band. (See the main text about (**i**)–(**iii**)).

Whereas the apparent similarity of crystal structures of $CuGaO_2$ and ZnO would promise development of their hybrids, the formation of the p-n junction between $CuGaO_2$ and ZnO is not necessarily obvious because there are no atoms within the distance of 1.7188 (1.8764) Å of Cu (Zn) in the c-plane [see Figure S1a,b in the Electronic Supplementary Information (ESI)]. Moreover, the lattice mismatch of ~10% is quite large, so massively strained growth with a large transition regime between $CuGaO_2$ and ZnO can be anticipated. Thus, further investigation is needed.

Raman spectroscopy is an effective tool for nondestructive characterization of lattice dynamics for various materials and can be used to obtain information on the crystal structures, lattice defects, and phase transitions via the changes in lattice vibration [19]. For example, the half-width of a Raman peak is associated with the crystallinity [20]; a shift in the peak position indicates the strain in the crystal lattice [21], and the covalency of bonds [22], and the intensity of the peak reflects the physical properties such as the concentrations of components [23]. Cu-based delafossite materials are being actively studied by Raman techniques [24,25], and many discussions arise from them. For example, impurity phases undetectable by X-ray techniques, such as CuO [26] and $Cu_2O$ [27], were observed, and the identification of many Raman signals that could not be assigned to those of the ideal $CuGaO_2$ structure [28] was suggested in Raman spectra of $CuGaO_2$ crystals.

To characterize the heterostructures, X-ray photoelectron (XP) spectroscopy (XPS) may provide insights into the chemical changes in the constituents of materials. Because XPS uses X-rays as an excitation source, the excitation light does not damage the substances and thus XPS can be used to measure various materials such as polymers, metallic materials, and insulators. X-rays from Mg K$\alpha$ and Al K$\alpha$ sources are often used to irradiate materials. They supplement the photoelectrons emitted by the ionization of the materials so that energy analysis can be performed. When one element bonds with another, the intra-atomic electronic state changes, and the binding energy of the core level (CL) also changes; thus, an XP peak shift can be observed. These changes are regarded as chemical shifts and are the most important features of the XP spectra, as they enable state analysis. Gao et al. [29] recently investigated Ca-doped $CuScO_2$ through XPS and reported a possible charge balance in the $CuScO_2$ films owing to the formation of copper vacancy defects after Ca

doping. A delafossite thin film on a polymethyl methacrylate or $SiO_2/Si$ substrate was also studied via XPS and its electrical and optical properties have been reported [30,31]. In addition to the above-mentioned basic research, recent works expanded the application scope of delafossite materials and heterojunctions. Especially, "self-powered solar-blind photodetectors" are one of the promising devices owing to the increasing demand for energy saving, miniaturization, and high efficiency [32–34].

In our previous work [9], the hydrothermal synthesis of a hexagonal platelet crystal of $CuGaO_2$ with a delafossite structure was reported, and it was demonstrated that the particle size of $CuGaO_2$ could be controlled by adding polyethylene glycol (PEG) with different molecular weights ($M_W = 6000$ and $20,000$). The photocatalytic activity of $CuGaO_2/ZnO$ hybrids with a particle size of approximately 8 μm has been reported. The results showed that the hybrids had better photocatalytic properties than $CuGaO_2$ or ZnO alone, where efficient electron-hole separation in the heterojunction with a ZnO layer on the *c* plane of the delafossite crystal was suggested as a driving force for the catalytic activity. In this study, XRD and confocal micro-Raman spectra of $CuGaO_2$ platelet crystals and $CuGaO_2/ZnO$ hybrids were measured to determine their structures and especially for micro-Raman revealed the vibrational properties of the targeted particle within several μm laser spot size. The changes in the valence states and binding energies of the synthesized $CuGaO_2$ and $CuGaO_2/ZnO$ hybrids were also examined by the XPS.

## 2. Materials and Method

### 2.1. Preparation of CuGaO₂ Powder

The following reagents were used in the hydrothermal synthesis of delafossite $CuGaO_2$: $Cu(NO_3)_2 \cdot 2.5H_2O$ (Sigma-Aldrich, St. Louis, MO, USA; 99%+), $Ga(NO_3)_2 \cdot 8H_2O$ (Nacalai Tesque Co., Kyoto, Japan), ethylene glycol (EG) (Kishida Chem. Co., Osaka, Japan), PEG 6000 ($M_W = 6000$) (Kishida Chem. Co.), PEG 20,000 ($M_W = 20,000$) (Kishida Chem. Co.), and KOH (Kishida Chem. Co.). Two samples of $CuGaO_2$ were synthesized using PEG 6000 and PEG 20,000, and the pH of the precursor solution was adjusted accordingly.

The synthesis method has been described in detail elsewhere [9]. Briefly, $Cu(NO_3)_2 \cdot 2.5H_2O$ (1 mmol) and $Ga(NO_3)_2 \cdot 8H_2O$ (1 mmol) were dissolved together in 3.6 mL of deionized water. Next, 3 mL EG and 0.1 g of PEG 6000 or PEG 20,000 were added. $KOH_{aq}$ (5 mmol) was introduced to each solution to adjust the pH to approximately 8.5. Each obtained deep-blue solution was poured into a Teflon-lined autoclave vessel, which was placed in an oven at 190 °C [35]. After a reaction time of 56 h, it was naturally cooled to room temperature. The precipitate was filtered and washed with a dilute ammonia (0.1 N) and nitrate (0.1 N) solution twice, respectively. A brown powder of $CuGaO_2$ platelets was obtained. Hereafter, $CuGaO_2$ samples synthesized with PEG 6000 and PEG 20,000 are denoted as CGO1 and CGO2, respectively. To transfer it to the next hybrid synthesis, as-prepared $CuGaO_2$ was annealed in air at 400 °C for 2 h to remove remaining organic entities. It was confirmed that $CuGaO_2$ remained stable even after the annealing [9].

### 2.2. Preparation of CuGaO₂/ZnO Hybrids

To prepare $CuGaO_2/ZnO$ hybrids, $Zn(CH_3COO)_2 \cdot 2H_2O$ (Kishida Chem. Co.) was used as the starting material for ZnO. As shown in Table 1, various quantities of $Zn(CH_3COO)_2 \cdot 2H_2O$ were added to 15 mL of deionized water to achieve different [Zn]/[Cu] ratios. The ammonia reagent (28%, Kishida Chem. Co.) was added to adjust the pH to approximately 7, and the obtained zinc precursor solution was poured into a Teflon vessel with the $CuGaO_2$ powder (CGO1 or CGO2) annealed for 2 h at 400 °C, as mentioned above. After hydrothermal treatment for 6 h in an oven at 180 °C in a Teflon vessel sealed in a stainless autoclave, powdered $CuGaO_2/ZnO$ samples were dried at 60 °C for 2 h. The hybrid samples are labeled as CZ1 ([Zn]/[Cu] = 1.65 mmol/0.6 mmol = 2.75); CZ2 ([Zn]/[Cu] = 3.3 mmol/0.6 mmol = 5.5); CZ3 ([Zn]/[Cu] = 6.6 mmol/0.6 mmol = 11), which was synthesized with CGO1; and CZ4 ([Zn]/[Cu] = 9.9 mmol/0.6 mmol = 16.5), which was

synthesized with CGO2. (See Table 1) All of the as-synthesized CZ samples were annealed under a reducing atmosphere of $H_2/N_2$ (5%/95%) at 400 °C for 10 h.

**Table 1.** Sample names and synthesis conditions for $CuGaO_2/ZnO$ hybrids. (ac = $CH_3COO$).

| Sample Name | CuGaO$_2$ Used | Mass of Zn(ac)$_2$·2H$_2$O | Nominal [Zn]/[Cu] Ratio |
|:---:|:---:|:---:|:---:|
| CZ1 | CGO1 | 0.18 g | 2.75 |
| CZ2 | CGO1 | 0.36 g | 5.5 |
| CZ3 | CGO1 | 0.72 g | 11 |
| CZ4 | CGO2 | 1.08 g | 16.5 |

*2.3. Characterization*

The synthesized $CuGaO_2/ZnO$ hybrids were examined by X-ray diffraction (XRD) analysis (PANalytical X'pert Pro MPD) and scanning electron microscopy (SEM; JEOL, JSM-6010LA). The simulated XRD patterns of the reference crystals were obtained using the RIETAN-FP program [36]. Raman scattering experiments were conducted using an InVia Raman spectrophotometer (Renishaw) in confocal mode. XP spectra were recorded on a PHI5000 VersaProbe X-ray photoelectron spectrometer with an Al K$\alpha$ X-ray source (ULVAC-PHI). The energies were calibrated with C1s peak (285 eV). The XPS CL spectra were baseline corrected with a Shirley background, and a least square fitting was conducted for the respective CL spectrum, using Voigt functions with a Gaussian component width (Gw) and Lorentzian component width (Lw) by Igor Pro 8.0 software. The shape parameter was given by Lw/Gw.

## 3. Results and Discussion

*3.1. Structural Analysis of CuGaO$_2$ and CuGaO$_2$/ZnO Hybrids*

The structure and morphology of the interfacial $CuGaO_2/ZnO$ hybrids were investigated by powder XRD and SEM. The XRD patterns of the hybrids after $H_2/N_2$ annealing are shown in Figure 3, together with those of the CGO1 and CGO2 base crystals [9]. The simulated patterns of $CuGaO_2$ (ICDD PDF 01-082-8561 for 3R and ICDD PDF 04-011-1001 for 2H) and ZnO (ICDD PDF 04-003-2106) are also shown for comparison. From the figures of CGO1 and CGO2, it is elucidated that hydrothermal synthesis resulted in the formation of rhombohedral (3R) $CuGaO_2$ as a single phase (signified by "D" in Figure 3). The hydrothermal deposition of ZnO on $CuGaO_2$ generated $CuGaO_2/ZnO$ hybrids. As shown in Figure 3, new XRD peaks in CZ1–4 are attributed to wurtzite-type ZnO (indicated by the # symbol). Contrary to the early work [9], a post-reduction process was employed here, i.e., the CZ1-4 samples were heat-treated in a hydrogen atmosphere to induce more carriers in n-type ZnO [37]. For the CZ1–3 samples, where CGO1 was used as a base crystal, the X-ray reflection peaks (#) are superimposed on the peaks (D) corresponding to 3R $CuGaO_2$, and the intensity increases with the amount of $Zn(CH_3COO)_2 \cdot 2H_2O$ involved in the reaction (see Table 1).

While the XRD peaks of the CZ1 and CZ2 samples are consistent with the results of our previous work [9], the CZ3 sample exceptionally exhibits a small peak at 30.97° (†), which indicates the formation of spinel-type cubic $CuGa_2O_4$ (*Fd-3m*) (ICDD PDF 04-001-9116) [38]. There are also three additional peaks (*) in CZ3, which are not assigned to any single component of copper oxide ($Cu_2O/CuO$) or gallium oxide ($Ga_2O_3$) but might be attributed to spinel-type tetragonal $ZnCu_2O_4$ (*I4$_1$/amd*) [39–41]. (Figure S2 in the ESI.) $ZnCu_2O_4$ spinel is not available in nature but has recently been revealed by computational predictions [42–44]. This evolution could be specifically promoted during the simultaneous production of $CuGa_2O_4$ spinel, as hypothesized by a sequential reaction scheme shown in Figure 4. Before the formation of $CuGaO_2/ZnO$ hybrid (CZ3), $CuGaO_2$ partially decomposes in the annealing process in air to CuO and $Ga_2O_3$, as seen in **Reaction 1** in Figure 4 (confirmed by the Raman and XPS data below). According to Hautier et al. [40], the development of $ZnCu_2O_4$ requires a more oxidizing environment. In our case, additional oxygen might be supplied when water molecules were adsorbed on the species

during the hydrothermal synthesis of ZnO. If this is the case, partial formation of $ZnCu_2O_4$ and $CuGa_2O_4$ spinels could occur in the post-reduction process (**Reaction 2** in Figure 4), because the XRD patterns of the hybrids before the $H_2/N_2$ reduction exhibited only ZnO and $CuGaO_2$ without any additional phases such as $ZnCu_2O_4$ and $CuGa_2O_4$, as shown in Figure S3 in the ESI.

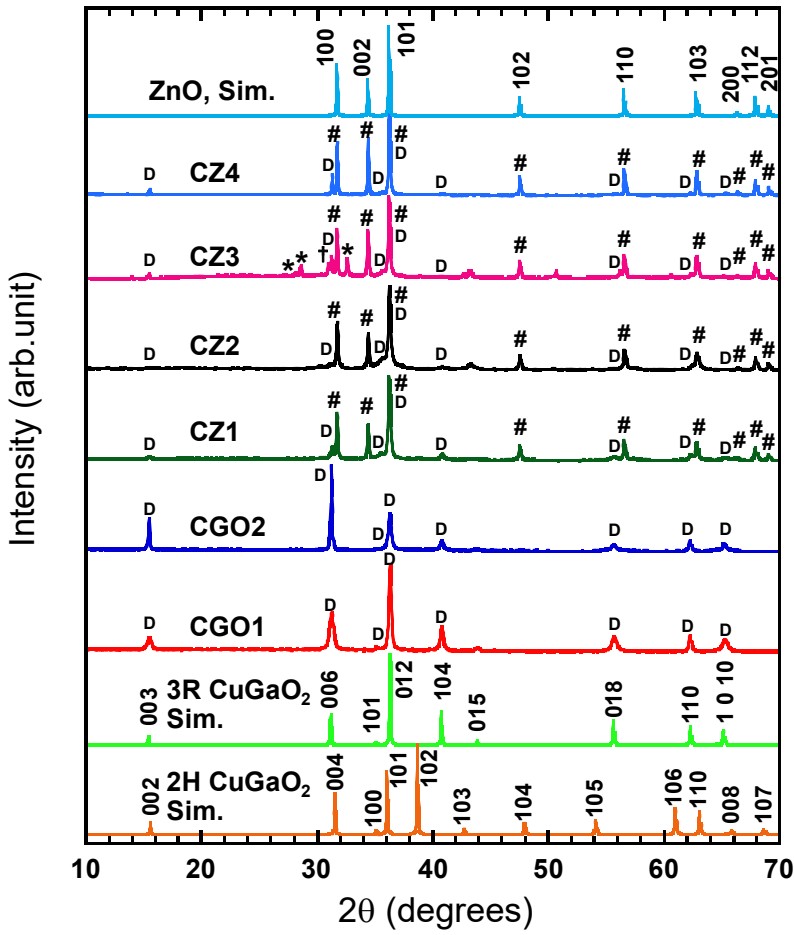

**Figure 3.** XRD patterns of the $CuGaO_2$/ZnO hybrid samples post-annealed in $H_2/N_2$, together with those of the $CuGaO_2$ base crystals (CGO1 and CGO2) for comparison [9]. The simulated patterns of 3R and 2H $CuGaO_2$ and ZnO are also shown. The symbols D, #, ǂ, and * indicate 3R $CuGaO_2$, ZnO, $CuGa_2O_4$, and $ZnCu_2O_4$, respectively. (See the main text for details.).

$$4CuGaO_2 + O_2 \rightarrow 4CuO + 2Ga_2O_3 \qquad \textbf{(Reaction 1)}$$

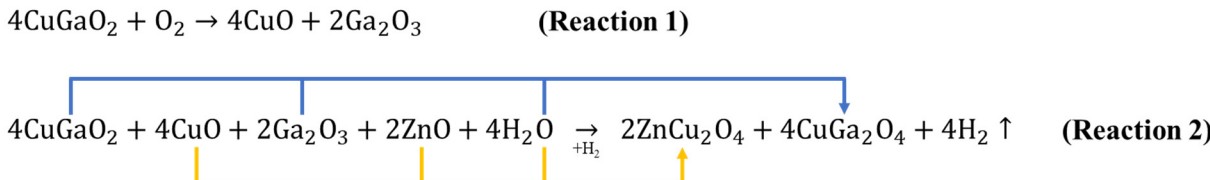

$$4CuGaO_2 + 4CuO + 2Ga_2O_3 + 2ZnO + 4H_2O \underset{+H_2}{\rightarrow} 2ZnCu_2O_4 + 4CuGa_2O_4 + 4H_2 \uparrow \qquad \textbf{(Reaction 2)}$$

**Figure 4.** Reactions for possible formations of $ZnCu_2O_4$ and $CuGa_2O_4$.

To look at the CZ4 prepared using CGO2, the sharp (006) peak of $CuGaO_2$ is visible in the XRD pattern, and its relative intensity is high than that of the (104) peak in the same crystal ($I_{006}/I_{104} = 13.1$, which is closer to $I_{006}/I_{104} = 7.57$ for CGO2 than $I_{006}/I_{104} = 1.57$ for CGO1), suggesting the effective deposition of ZnO on the *c* plane (006) of the $CuGaO_2$ platelets in CZ4. (See Figure S4 in the ESI.)

Small peaks at ~43°, ~50°, and ~61° are distinct in the XRD patterns of CZ2 and CZ3 (Figure 3) but are absent in the XRD patterns of the same hybrids before the post-reduction process [9]. From a database survey, they were assignable to $Cu_2O$ and metallic

Cu with face-centered cubic and hexagonal structures, respectively, as shown in Figure S2, and Table S1 in the ESI.

### 3.2. SEM Observation of CuGaO$_2$ and CuGaO$_2$/ZnO Hybrids

Figure 5 shows the SEM images of CuGaO$_2$ and CuGaO$_2$/ZnO hybrids samples. The CuGaO$_2$ base crystals exhibit well-defined and characteristic morphologies of hexagonal plates with sizes of 1–3 μm [CGO1, Figure 5a] and 5–8 μm [CGO2, Figure 5b]. The variations in size are related to the molecular weight of the PEG used in the synthesis [9]. For CGO2 when PEG 20,000 was used, the XRD peak corresponding to the (006) plane was relatively strong, indicating an enhanced growth of hexagonal plate-like particles. Because more –C–O–C– moieties are available in the PEG with higher molecular weight, more adsorption occurs on the (006) plane, and the growth of the *c* plane is promoted [9].

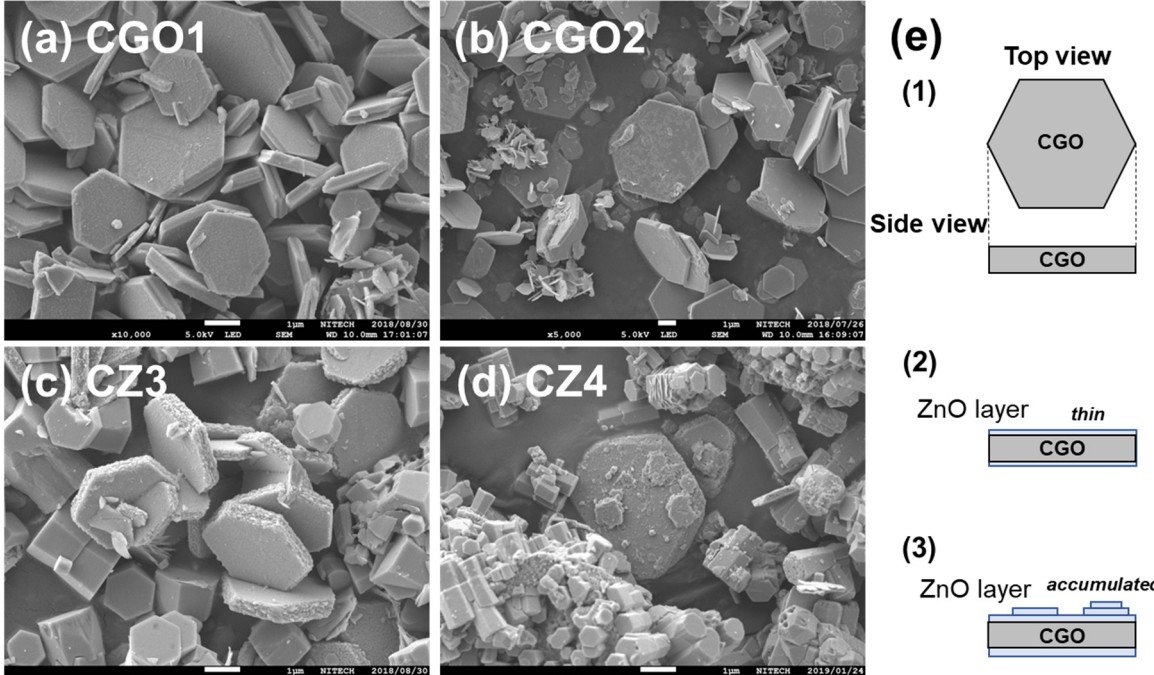

**Figure 5.** SEM images of the CuGaO$_2$ base crystals (**a**,**b**) and CuGaO$_2$/ZnO hybrids (**c**,**d**). CGO1 (**a**) was synthesized with PEG 6000 while CGO2 (**b**) with PEG 20,000. CZ3 (**c**) and CZ4 (**d**) were hybridized with CGO1 and COG2, respectively. The scale bar shows 1 μm. [Zn]/[Cu] ratio was adjusted for CZ3 and CZ4, as given in Table 1. (See the details in the Section 2). SEM images of the CZ1 and CZ2 samples can be found in Figure S5 in the ESI. (**e**) Schematical illustration of ZnO layer on CuGaO$_2$(CGO). (**1**) bare CGO in top and side views, (**2**) CGO with thin ZnO layers, and (**3**) CGO with thicker, rather accumulated ZnO layers in side view.

SEM investigation of the hybrids synthesized with various amounts of Zn(CH$_3$COO)$_2$·2H$_2$O elucidates that the morphological results of the samples are dependent on the starting base crystals. Here, SEM images of CZ3 ([Zn]/[Cu] = 11) and CZ4 ([Zn]/[Cu] = 16.5) are shown in Figure 5c,d. (SEM images of the CZ1 and CZ2 samples are given in Figure S5 in the ESI.) Unlike CZ1 and CZ2, the surfaces of the plate-like CuGaO$_2$ particles in CZ3 (Figure 5c) are completely covered with a ZnO layer. As for CZ4 (Figure 5d), which was hybridized with the larger CGO2 base crystal, the ZnO well covers the hexagonal CuGaO$_2$ particles to form a thick layer, and individual ZnO hexagonal blocks are also observed. A variety of ZnO layering on the CuGaO$_2$ plate is illustrated in Figure 5e. SEM-EDS results for the hybrids, shown in Figures S6–S9 in the ESI, indeed reveal the presence of ZnO on the CuGaO$_2$ platelets [45].

### 3.3. Micro-Raman Investigation

Micro-Raman spectra of CGO1, CGO2, CZ3, and CZ4 are shown in Figure 6a. A Raman microscope was used for the measurement and a hexagonal platelet was imaged with a 100× objective. The optical pictures recorded during the spectral measurements are shown in Figure 6b,c. The laser wavelength was 532 nm, and the depth of the irradiated spot was approximately 1 μm because the measurement was performed in the confocal mode (See Figure 6d–f) [46]. The beam spot size was estimated to be ~2 μm$\phi$, which was small enough to detect the Raman signal of a single $CuGaO_2$ (size 3~8 μm) and $CuGaO_2$/ZnO plates. (Figure 6d) As shown in Figure 6a, Raman signals corresponding to $CuGaO_2$ [28] were obtained in CGO1, CGO2, and CZ3, and a signal attributed to ZnO was confirmed [47] in CZ3 and CZ4.

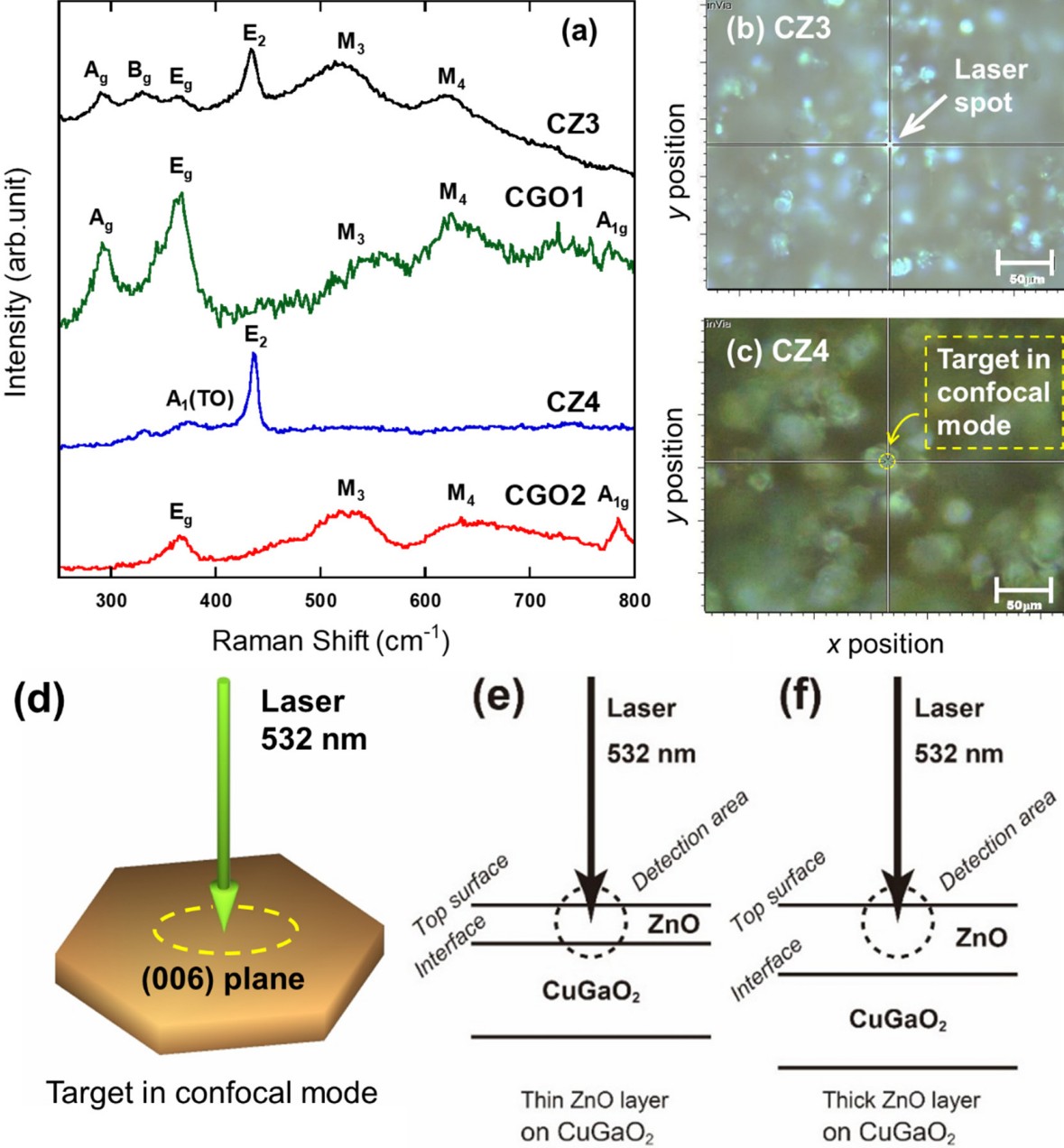

**Figure 6.** (**a**) Raman spectra of $CuGaO_2$ and $CuGaO_2$/ZnO, and optical microscopy images of (**b**) CZ3 and (**c**) CZ4. The scale bar is 50 μm. (**d**) Illustration of micro-Raman experiment and (**e**,**f**) schematic drawings of depth region for the confocal Raman detection.

Looking closer, Raman spectra of CGO1 and CGO2 exhibit peaks corresponding to the $E_g$, $M_3$, $M_4$, and $A_{1g}$ modes of delafossite $CuGaO_2$ crystals [28], observed at approximately 380, 521, 610, and 780 cm$^{-1}$, respectively. According to the literature [28], the modes labeled as $M_3$ and $M_4$ are attributed to the stress-induced modes of $A_g$ (and/or $B_u$) and $B_u$ at the X point, respectively, meaning that our observations may be specific to the hydrothermal synthesis of $CuGaO_2$. For CGO1 and CZ3, Raman peaks with lower wavenumbers are very similar to the vibrational modes of CuO ($A_g$ and $B_g$) [48], suggesting the partial decomposition of $CuGaO_2$, as discussed with **Reaction 1** in Figure 4. Moreover, the observed $CuGaO_2$ peaks in the CZ3 sample shift slightly to lower wavenumbers. This shift is caused by the distortion of the $CuGaO_2$ crystal induced by strains at the hetero-interface between $CuGaO_2$ and ZnO (see the discussion on XPS results shown later). In the Raman spectra of CZ3 and CZ4, a signal at 435 cm$^{-1}$ is observed that was correlated to the $E_2$ mode of ZnO on the $CuGaO_2$ plate. In CZ4, a small peak at ~380 cm$^{-1}$ emerged and was assigned to the $A_1$(TO) mode of ZnO [47,49]. These ZnO-related modes confirm that $CuGaO_2$ was successfully hybridized with ZnO. As for the CZ3 sample, Raman signals from the $CuGaO_2$ plate are also detected in addition to the ZnO peaks, because the ZnO layer is less than 1 μm thickness (See Figure 5e(2) and Figure 6e). Contrarily, CZ4 has thicker, accumulated ZnO layers (See Figure 5e(3)), and no vibrational modes of $CuGaO_2$ are visible in the Raman spectrum apart from a strong ZnO peak (See Figure 6f). Many isolated ZnO blocks appear around the hybrid because the highest $Zn(CH_3COO)_2 \cdot 2H_2O$ content was used ([Zn]/[Cu] = 16.5) in CZ4, which is in agreement with the XRD results. A comparison of the SEM images of CZ3 and CZ4 confirms that small hexagonal blocks of ZnO are formed and accumulate on the surface of the CGO2 plates in CZ4, while CZ3 has a thinner ZnO coating on the CGO1 particles. In our previous study, we found that CZ4 had better photocatalytic properties because the larger $CuGaO_2$ hexagonal plates were well-coated with ZnO [45]. Hence, a sufficient thickness of the ZnO layer would be important to extract electrons from electron-hole pairs generated at the p-n interface between $CuGaO_2$ and ZnO and prevent electrons from returning to the interface with p-type $CuGaO_2$, resulting in the higher photocatalytic performance, as demonstrated in our previous study.

*3.4. Unique Reaction in the Boundary between $CuGaO_2$ and ZnO in CZ3*

As seen in Section 3.1, the CZ3 samples experienced the crystallization of two spinel phases. Here, the formation mechanism of $ZnCu_2O_4$ and $CuGa_2O_4$ shall be discussed. Firstly, to be mentioned, the particle size of CGO has a significant impact on the formation of $ZnCu_2O_4$ and $CuGa_2O_4$; CGO1 and CGO2 were prepared using PEG 6000 and PEG 20,000, respectively. Compared with CGO2, CGO1 has a smaller size, indicating that it has a higher specific surface area [45]. Therefore, more oxygen molecules are likely to be adsorbed on CGO1 and more dominantly trigger the decomposition reaction. For the CZ3 sample synthesized with [Zn]/[Cu] ratio = 11 and CGO1 used, CuO and $Ga_2O_3$ produced by the partial decomposition may react with excess ZnO to develop $ZnCu_2O_4$ and $CuGa_2O_4$ during the post-reduction process according to **Reactions 1 and 2**, as shown in Figure 4. In comparison, when [Zn]/[Cu] ratio maintains as 11 and CGO2 is used instead of CGO1, the XRD pattern (Figure S10 in the ESI) only shows peaks of ZnO and $CuGaO_2$, while those attributed to $ZnCu_2O_4$ and $CuGa_2O_4$ are not detected. It indicates that increasing the particle size of $CuGaO_2$ significantly reduces the number of oxygen molecules adsorbed on the hexagonal platelets for partial decomposition, which leads to the production of small amounts of $ZnCu_2O_4$ and $CuGa_2O_4$ phases in the subsequent reduction process and leads to more efficient hybridization with ZnO.

The spinel synthesis in $CuGaO_2$/ZnO hybrid is itself interesting. However, to take into consideration the results of XRD, SEM, and Raman investigations, a further experiment of XPS is determined to be performed for the CZ4 and pristine CGO2 base crystal because the CZ sample possesses the sufficient ZnO layer without any additional phases like CuO and the spinels suspected to hinder the carrier separation in the boundary between $CuGaO_2$ and ZnO.

### 3.5. XPS Analysis for the CuGaO$_2$ Base Crystal and CuGaO$_2$/ZnO Hybrid

A broad scan XPS spectrum is obtained to identify the elements in the respective sample. Figure 7 shows the comparison of the wide-scan spectra of CGO2 and CZ4. The photoelectron peaks of the main constituents, Cu, Ga, Zn, and O and Auger Cu, Ga, Zn LMM, and O KLL peaks are observed, wherein Cu$_{2p_{3/2}}$ and Cu$_{2p_{1/2}}$ were detected at 932 and 952 ev [50], respectively; Ga$_{2p_{3/2}}$ and Ga$_{2p_{1/2}}$ at 1117 and 1144 eV [51], respectively, and O$_{1s}$ at approximately 530 eV [52]. The XPS peaks of Zn were detected only for CZ4 and were assigned to Zn$_{2p_{3/2}}$ and Zn$_{2p_{1/2}}$ (1021.5 and 1044.5 eV, respectively) [53,54]. The positions and width of the detected XPS peaks were tabled in Table 2.

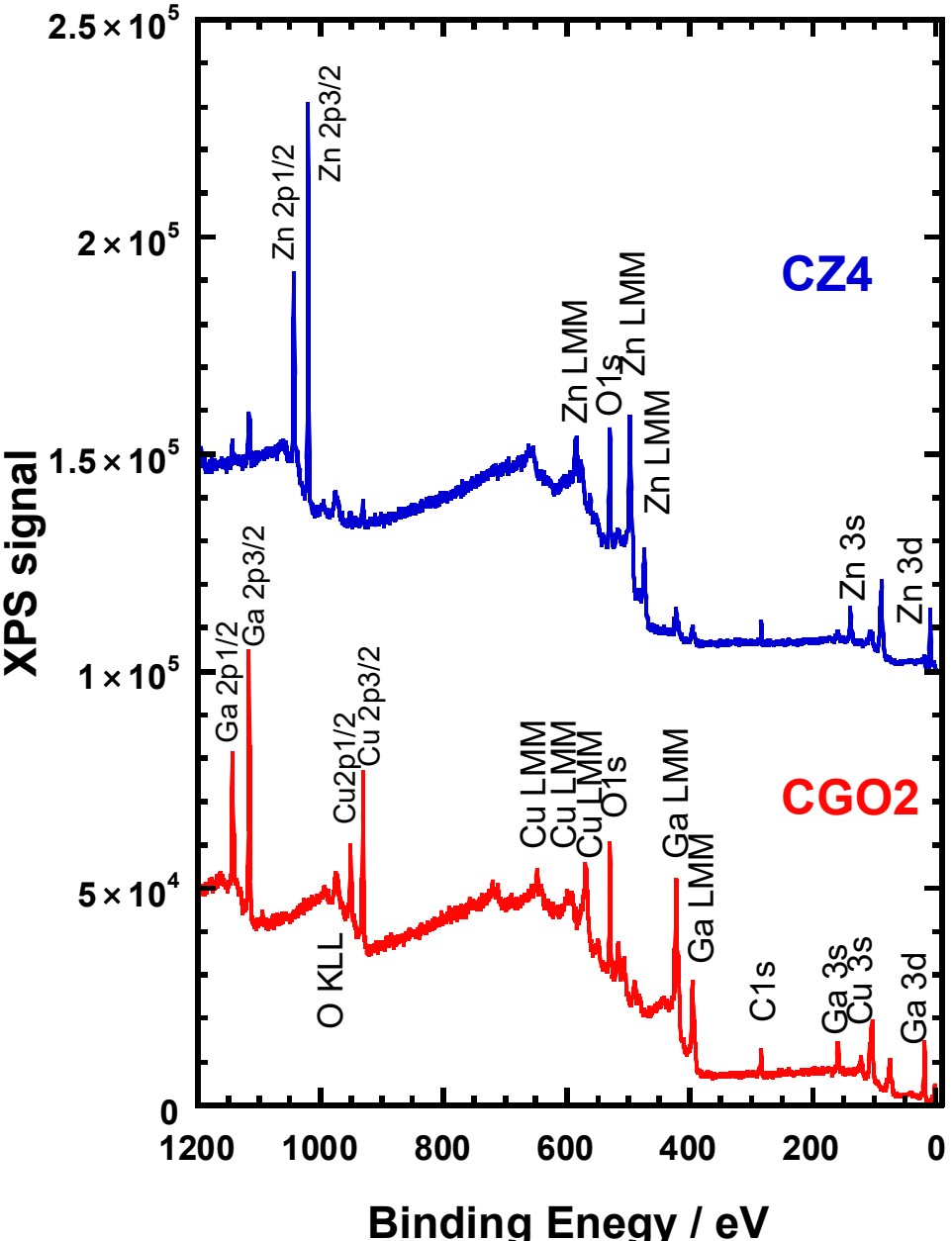

**Figure 7.** The survey scan spectra of the CGO2 base crystal and CZ4 hybrid.

**Table 2.** Peak position and width ($\pm 0.1$ eV) of each of the XPS CL peaks for CGO2 and CZ4.

| Sample | XPS CL | Peak Position/eV | Width/eV |
|--------|--------|------------------|----------|
| CGO2 | $Cu_{2p_{3/2}}$ | 932.5 | 1.2 |
|  | $Cu_{2p_{1/2}}$ | 952.3 | 1.4 |
|  | $Ga_{2p_{3/2}}$ | 1117.8 | 1.7 |
|  | $Ga_{2p_{1/2}}$ | 1144.7 | 1.6 |
|  | $O_{1s}$ | 530.3 | 1.5 |
| CZ4 | $Cu_{2p_{3/2}}$ | 932.69 | 2.4 |
|  | $Cu_{2p_{1/2}}$ | 952.6 | 2.3 |
|  | $Ga_{2p_{3/2}}$ | 1117.8 | 2.3 |
|  | $Ga_{2p_{1/2}}$ | 1144.8 | 2.4 |
|  | $Zn_{2p_{3/2}}$ | 1021.7 | 1.7 |
|  | $Zn_{2p_{1/2}}$ | 1044.8 | 1.8 |
|  | $O_{1s}$ | 530.3 | 1.8 |

Figures 8a and 9a show the $Cu_{2p_{3/2}}$ XPS CL spectra of CGO2 and CZ4, respectively. After subtracting the Shirley background, the experimental peak at 932.5 eV was fitted with Voigt functions using Igor Pro 8.0 software. At first, the chemical state of Cu in CGO2 was analyzed in detail by investigating $Cu_{2p_{3/2}}$ XPS CL signal [55]. Although it seemed possible to be fitted with a single Voigt function peaked at 932.5 eV assignable to $Cu^+$ ions, the addition of the second Voigt function at 933.6 eV for $Cu^{2+}$ state provided a more reliable fitting result, which can elucidate the influence of $Cu^{2+}$ ions in this substance. As seen in Figure 8a, the ratio of $Cu^{2+}$ ions is low enough ~4%. On the other hand, the Cu XPS signal for CZ4 in Figure 9a is found to be very broadened (full width at half maximum (FWHM) ~2.4 eV) in comparison with that of CGO2 (FWHM ~1.2 eV), and well fitted with two Voigt functions peaked at 932.4 eV for $Cu^+$ and 933.6 eV for $Cu^{2+}$ ions. The analytical data are listed in Table 3. The feature was not varied after surface etching by argon ion beam sputtering. The equivalent amount of $Cu^{2+}$ to $Cu^+$ ions is detected for the CZ4 sample.

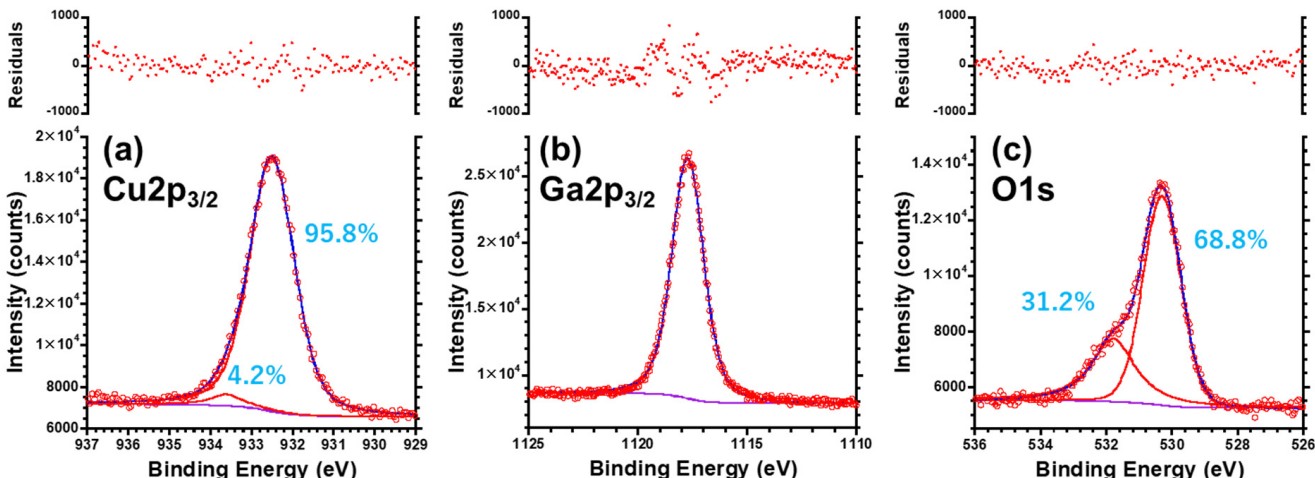

**Figure 8.** XPS spectra of (**a**) $Cu_{2p_{3/2}}$, (**b**) $Ga_{2p_{3/2}}$, and (**c**) $O_{1s}$ CLs for the CGO2 base crystal and fitting results using Voigt functions. The fitting results are summarized in Table 3.

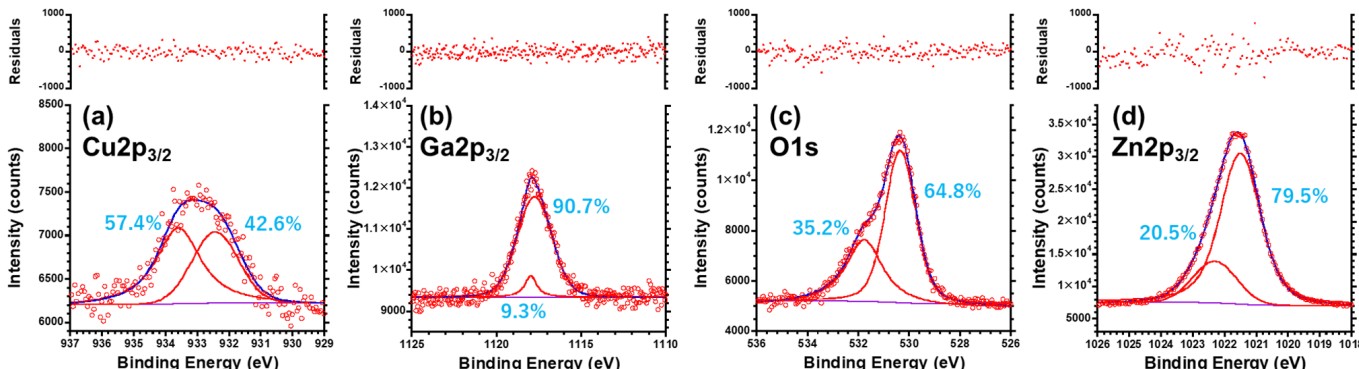

**Figure 9.** XPS spectra of (**a**) $Cu_{2p_{3/2}}$, (**b**) $Ga_{2p_{3/2}}$, (**c**) $O_{1s}$, and (**d**) $Zn_{2p_{3/2}}$ CLs for the CZ hybrid sample and fitting results using Voigt functions. The fitting results are summarized in Table 3.

**Table 3.** Fitting results (peak position, area, and width) for $Cu_{2p_{3/2}}$, $Ga_{2p_{3/2}}$, $O_{1s}$, and $Zn_{2p_{3/2}}$ XPS CL signals with one or two Voigt functions about the CGO2 and CZ4 samples. The shape parameter is defined by Lw/Gw (Gw: Gaussian width component (%), Lw: Lorentzian width component (%)).

| | | Peak Position/eV | | Area (%) | | Width/eV | | Shape (=Lw/Gw) | Gw (%) | Lw (%) |
|---|---|---|---|---|---|---|---|---|---|---|
| **CGO2** | | | | | | | | | | |
| $Cu_{2p_{3/2}}$ | $Cu^+$ | 932.51 | ±0.01 | 95.8 | ±1.5 | 1.28 | ±0.05 | 0.46 | 68.5 | 31.5 |
| | $Cu^{2+}$ | 933.60 | Fixed | 4.2 | ±1.1 | 0.97 | ±0.22 | $8.0 \times 10^3$ | 0.0 | 100.0 |
| $Ga_{2p_{3/2}}$ | Ga#1 | 1117.74 | ±0.01 | 100.0 | - | 1.71 | ±0.05 | 0.50 | 66.3 | 33.7 |
| $O_{1s}$ | O#1 | 530.32 | ±0.01 | 68.8 | ±1.7 | 1.35 | ±0.04 | 0.10 | 90.9 | 9.1 |
| | O#2 | 531.80 | ±0.03 | 31.2 | ±2.4 | 1.55 | ±0.36 | 4.41 | 18.5 | 81.5 |
| **CZ4** | | | | | | | | | | |
| $Cu_{2p_{3/2}}$ | $Cu^+$ | 932.44 | ±0.22 | 42.6 | ±14.5 | 1.72 | ±0.82 | 0.00 | 99.9 | 0.1 |
| | $Cu^{2+}$ | 933.60 | ±0.18 | 58.4 | ±16.1 | 1.71 | ±1.09 | 4.24 | 16.4 | 81.6 |
| $Ga_{2p_{3/2}}$ | Ga#1 | 1117.76 | ±0.02 | 90.7 | ±9.5 | 2.41 | ±0.34 | 0.19 | 84.0 | 16.0 |
| | Ga#2 | 1117.98 | ±0.05 | 9.3 | ±7.7 | 0.84 | ±0.39 | $1.66 \times 10^3$ | 0.1 | 98.9 |
| $O_{1s}$ | O#1 | 530.36 | ±0.01 | 64.8 | ±2.1 | 1.40 | ±0.11 | 0.33 | 75.4 | 24.6 |
| | O#2 | 531.78 | ±0.04 | 35.2 | ±3.0 | 1.77 | ±0.33 | 37.3 | 2.6 | 97.4 |
| $Zn_{2p_{3/2}}$ | Zn#1 | 1021.51 | ±0.04 | 79.5 | ±10.3 | 1.49 | ±0.14 | 0.66 | 60.4 | 39.6 |
| | Zn#2 | 1022.31 | ±0.29 | 20.5 | 10.2 | 1.64 | ±0.39 | $3.8 \times 10^{-4}$ | 100.0 | 0.0 |

The copper valence state of $CuGaO_2$ base crystal was also estimated by modified Auger parameter $\alpha'$, defined by the following equation [56]:

$$\alpha' = \alpha + h\nu = \left( KE_{Auger\ electron} - KE_{photoelectron} \right) + h\nu$$
$$= KE_{Auger\ electron} + BE_{photoelectron} \tag{1}$$

where $h\nu$ is the photon energy of the exciting radiation, *KEs* are the kinetic energies of Auger electron and photoelectron, and *BE* is the binding energy of the photoelectron from the CL level of the targeted element. The Cu $L_3M_{4,5}M_{4,5}$ peak is detected at 916.63 eV for CGO2 (the magnified figure is given in Figure S1a in the ESI), and the Auger parameter is estimated to be 1849.14 eV. Table 4 shows the XPS CL peak location and $\alpha'$ for CGO2, compared with literature data of various copper compounds. They are found to be well matched to those of $Cu_2O$ [55,57,58], resulting in that the CGO2 was composed of well-defined $Cu^+$ monovalent ions. As for CZ4, unfortunately, Auger peak Cu LMM was not available because of its weakness and/or broadening of the corresponding peak. Nevertheless, the binding energy of the XPS CL signal sufficiently suggests the presence of $Cu^{2+}$ ions in the boundary between $CuGaO_2$ and ZnO, resulting from the annealing in air before the formation of a structural hybrid for $CuGaO_2$ and ZnO or an electronic potential slope in the interface region between $CuGaO_2$ and ZnO, inducing the carrier separation as expected.

**Table 4.** Binding energies for the $Cu_{2p_{3/2}}$ XPS CL peaks and kinetic energies for the Cu $L_3M_{4.5}M_{4.5}$ Auger peaks, and the modified Auger parameters $\alpha'$ for CGO2 and various copper compounds in eV ($\pm 0.1$ eV).

| Substance | $Cu_{2p_{3/2}}$/eV | Cu LMM/eV | $\alpha'$/eV | References |
|-----------|-----------|-----------|--------------|------------|
| CGO2 | 932.5 | 916.6 | 1849.1 | This work |
| Cu(metal) | 932.7 | 918.4 | 1851.1 | [55] |
| $Cu_2O$ | 932.4 | 916.5 | 1848.9 | [55,57,58] |
| CuO | 933.6 | 917.8 | 1851.4 | [55,57] |
| $Cu(OH)_2$ | 934.7 | 916.2 | 1850.9 | [58] |

Regarding the Ga element of the CGO2 compound (Figure 8b), the $Ga_{2p_{3/2}}$ the peak at 1117 eV is fitted with a single Voigt function with a shape parameter, defined by a ratio of Lorentzian component width to Gaussian component width, of 0.5 (Gaussian:66.3%, Lorentzian:33.7%). (See Table 3) On the other hand, two Voigt functions are used for CZ4 to obtain a good fit (Figure 9b), which is composed of the main peak (90.7%) at 1117.76 eV with a small, relatively sharp peak at 1117.98 eV (9.3%). The small peak was also detected in the measurement of depth profiles of $Ga_{2p_{3/2}}$ XPS CL spectra for the $CuGaO_2$/ZnO hybrid, thus indicative of a specific state of Ga-O bonds in a boundary between $CuGaO_2$ and ZnO. A comparison of the present data with the binding energies of metallic Ga (1116 eV) [59,60] and $Ga_2O_3$ (1118 eV) [51,60] reveals that the observed XPS peaks cannot be attributed to the lower valence state of Ga and the Ga elements resultantly have to be in a trivalent state coordinated with oxygens. As shown in Figures 8a,b and 9a,b, the Cu and Ga XPS CL signals of the hybrid samples are quite small because of the formation of the ZnO layer on $CuGaO_2$. However, a meaningful deconvolution analysis can be performed. Interestingly, the analysis of the $Ga_{2p}$ peak of the hybrid shows a larger FWHM (~2.3 eV), significantly broadened in comparison with a sharp peak of ~1.7 eV width for the CGO2 sample (See Table 2), suggesting the formation of the $CuGaO_2$/ZnO hybrid. This means that, while the CGO2 base crystal was composed of distorted but crystallographically-regulated $GaO_6$ octahedra, the CZ4 hybrid possessed $GaO_6$ octahedra with strained chemical bonding states in the boundary region with ZnO. Conclusively, the sufficient ZnO coverage of $CuGaO_2$, as also seen in the SEM image, weakened the XPS signals, but the detected signals are sensitively reflected by chemical states of $GaO_6$ octahedra influenced by the formation of the heterostructure with ZnO.

Ga $L_3M_{4.5}M_{4.5}$ kinetic energies for CGO2 and CZ4 are given in Table 5 and Figure S11b in the ESI. The estimated Auger parameter $\alpha'$ values are 2208.7 eV (CGO2) and 2208.2 eV (CZ4), which are not yet reported in the literature so far, and found to be 28 eV higher than that of $Ga_2O_3$ (2180.4 eV) [60]. Since "*the change in the Auger parameter for a given element is equal to the change in polarization energy of the structure*" [56], this will be characteristic of p-type semiconductive $CuGaO_2$ crystal.

**Table 5.** Binding energies for the $Ga_{2p_{3/2}}$ XPS CL peaks and kinetic energies for the Ga $L_3M_{4.5}M_{4.5}$ Auger peaks, and the modified Auger parameters $\alpha'$ for CGO2 and various gallium materials in eV ($\pm 0.1$ eV).

| Substance | $Ga_{2p_{3/2}}$/eV | Ga LMM/eV | $\alpha'$/eV | References |
|-----------|-----------|-----------|--------------|------------|
| CGO2 | 1117.8 | 1090.9 | 2208.7 | this work |
| CZ4 | 1117.8 | 1090.3 | 2208.1 | this work |
| Ga(metal) | 1116.5 | 1068.0 | 2184.5 | [59,60] |
| $Ga_2O_3$ | 1117.8 | 1062.6 | 2180.4 | [51,60] |

Regarding the oxygen, the $O_{1s}$ peak is coherently deconvoluted into two Voigt functions, O#1 (530.3 eV) and O#2 (531.8 eV), and the $O_{1s}$ peak in $CuGaO_2$ [Figure 8c] is composed of 68.8% O#1 and 31.2% O#2. For the CZ4 hybrid [Figure 9c], the $O_{1s}$ is 64.8% O#1 and 35.2% O#2, and has the same trend. According to the literature, adsorbed $H_2O$ molecules

and OH moieties are related to XPS peaks at 532.8 and 531.7 eV, respectively [52,61]. Thus, the O#2 signals for CGO2 and CZ4 are attributed to OH moieties owing to the hydrothermal synthesis. On the other hand, the observed binding energy for O#1 in these samples should originate from O as the main framework in the $CuGaO_2$ and ZnO structures. However, the chemical states are different between $CuGaO_2$ and ZnO because ZnO layer has four-fold coordinated oxygen with Zn ($OZn_4$), whereas in $CuGaO_2$, oxygen has four-fold-coordination with Cu and three Ga atoms combined with the $GaO_6$ octahedron, that is, an $OCuGa_3$ tetrahedron is present [Figures S1a and S12 in the ESI]. The broad feature of the $O_{1s}$ peak did not enable the different environments to distinguish.

Zn XPS signals were observed only for CZ4. As shown in Figure 9d, the $Zn_{2p_{3/2}}$ peak at 1021.7 eV is deconvoluted into two Voigt functions of Zn#1 at 1021.5 eV by 79.5% and Zn#2 at 1022.3 eV by 20.5%. In comparison with literature data including Zn (1021.8 eV) [56,57], ZnO (1022.1 eV) [57,62], and CuZn (1021.5 eV) [57], the obtained binding energy of Zn#1 appears to show rather lower valence state for the Zn element of the CZ4 sample, except for Zn#2 corresponding to a divalent state ($Zn^{2+}$) of the individual ZnO bulky blocks seen in the SEM image (Figure 5d). However, this discrepancy can be elucidated by the analysis of the modified Auger parameter $\alpha'$. The Auger parameters for Zn element of various materials (Zn, ZnO, CuZn, and Al-doped ZnO(AZO)) are compared in Table 6 with the present data for CZ4 given from the data of Auger peak Zn $L_3M_{45}M_{45}$ (See Figure S11c). It is seen that the value of the CZ4 hybrid ($\alpha' = 2010.1$ eV) is equivalent to ZnO ($\alpha' = 2010.3$ eV) [57,62] and Al-doped ZnO ($\alpha' = 2009.4$ eV) [63] and rather distinguishable from those of metallic Zn and alloy CuZn ($\alpha' = 2013.9$ eV) [57,62]. The results will be explained by the formation of a structural hybrid of $CuGaO_2$ and ZnO, which would induce effective transport of electron carriers to the ZnO region from the hetero-interface between the ZnO layer and $CuGaO_2$ crystal, like $Cu_2O$/ZnO and CuO/ZnO hetero-interfaces (See Table 6) [64].

**Table 6.** Binding energies for the $Zn_{2p_{3/2}}$ XPS CL peaks and kinetic energies for the Zn $L_{4,5}M_{4,5}$ Auger peaks, and the modified Auger parameters $\alpha'$ for CGO2 and various gallium materials in eV ($\pm 0.1$ eV). (AZO: Al-doped ZnO).

| Substance | $Zn_{2p_{3/2}}$/eV | Zn LMM/eV | $\alpha'$/eV | References |
|---|---|---|---|---|
| CZ4 | 1021.7 | 988.4 | 2010.1 | this work |
| Zn(metal) | 1021.8 | 992.1 | 2013.9 | [56,57] |
| ZnO | 1022.1 | 989.4 | 2010.3 | [57,62] |
| CuZn(alloy) | 1021.5 | 992.4 | 2013.9 | [57] |
| AZO(4at%Al) | 1022.0 | 987.3 | 2009.4 | [63] |
| AZO(2at%Al) on $Cu_2O$ | 1021.8 | 988.2 | 2010.0 | [64] |
| AZO(2at%Al) on CuO | 1022.2 | 987.9 | 2010.1 | [64] |

In summary, the $CuGaO_2$/ZnO hybrid was formed with a ZnO layer on the $CuGaO_2$ hexagonal platelet particles. The analysis of Cu and Ga $2p_{3/2}$ XPS CL signals suggested the presence of $Cu^{2+}$ ions and strained $GaO_6$ octahedra in the boundary between $CuGaO_2$ and ZnO. The Zn $2p_{3/2}$ binding energy and modified Auger parameter unveiled a unique specification of the hybrid sample, indicating the possibility of $CuGaO_2$/ZnO hetero-interface as a p-n type catalyst.

## 4. Conclusions

Hydrothermally synthesized $CuGaO_2$/ZnO hybrids, which were formed with $CuGaO_2$ hexagonal-plate base crystal and ZnO layer, were investigated. XRD and SEM investigations confirmed the successful deposition of ZnO on the $CuGaO_2$ plate, and the subsequent reduction process induced an extraordinary reaction in the interface between $CuGaO_2$ and ZnO, which could be tuned out by varying the particle size of $CuGaO_2$ and [Zn]/[Cu] ratio. Micro-Raman observations confirmed the vibrational modes of ZnO in the $CuGaO_2$/ZnO hybrids. XPS profiles showed the presence of Cu, Ga, and O in the $CuGaO_2$ and of Zn

in the $CuGaO_2/ZnO$ hybrids in addition to the elements from the base crystal, and the quantitative analysis of valence states of Cu ions ($Cu^+$ and $Cu^{2+}$) was performed. The $Cu^{2+}$ ratio was characteristically higher in the hybrids than in the $CuGaO_2$ crystal. The Ga XP spectra indicated that the $GaO_6$ octahedra in the inner structure of the $CuGaO_2$ base crystal were crystallographically well-constructed, while more strains were involved in the hetero-interface between $CuGaO_2$ and ZnO. The electronic state of the $CuGaO_2/ZnO$ hybrid was found to be significantly influenced by the formation of the hetero-interface between $CuGaO_2$ and ZnO in the analysis of the Zn XPS CL binding energies and modified Auger parameters, indicating effective transport of electron carriers to the ZnO region from the hetero-interface. It was concluded that the hydrothermal approach for the hybrid compound synthesis was promising and could cut out a new path for the development of optoelectronic devices.

**Supplementary Materials:** The following supporting information can be downloaded at: https://www.mdpi.com/article/10.3390/ceramics5040048/s1. Figure S1 (a): Extracted crystal structure of rhombohedral $CuGaO_2$ (R-3m). Blue, red, and green balls show Cu, O, Ga elements, respectively. The lattice parameters are taken from ICDD PDF 01-082-8561; Figure S1 (b): Extracted crystal structure of wurtzite(hexagonal) ZnO ($P6_3mc$). Grey and red balls show Zn and O elements, respectively. The lattice parameters are taken from ICDD PDF 04-003-2106; Figure S2: XRD pattern of CZ3 sample (after the $H_2$ annealing process) and assignment of the corresponding crystals; Table S1: XRD peak assignment for the CZ3 sample; Figure S3: XRD patterns of $CuGaO_2/ZnO$ hybrids of CZ1-4 before the $H_2$ annealing process. (#:wurtzite ZnO, D:rhombohedral(3R) $CuGaO_2$); Figure S4: XRD pattern (magnified) of CZ4 sample after the $H_2$ annealing process; Figure S5: SEM images of the CZ1 (a) and CZ2 (b) samples; Figure S6: SEM-EDS results of the CZ1 sample; Figure S7: SEM-EDS results of the CZ2 sample; Figure S8: SEM-EDS results of the CZ3 sample; Figure S9: SEM-EDS results of the CZ4 sample, which are taken, for comparison with Figures S5–S7, from the literature, Ref. [45] (M. Choi, S. Yagi, Y. Ohta, K. Kido and T. Hayakawa, J. Phys. Chem. Solids, 150 (2021) 109845); Figure S10: Comparison of XRD patterns for two $CuGaO_2/ZnO$ hybris synthesized with the same [Zn]/[Cu] ratio (=11) but by use of the different base crystals, CGO1 and CGO2 (#:wurtzite ZnO, D:rhombohedral(3R) $CuGaO_2$); Figure S11: Auger $L_3M_{4,5}M_{4,5}$ spectra for CGO2 ((a) Cu LMM and (b) Ga LMM) and CZ4 ((b) Ga LMM and (c) Zn LMM); Figure S12: Overview of $OCuGa_3$ polyhedra in 3R $CuGaO_2$ structure.

**Author Contributions:** M.C.: investigation, writing the original draft, data curation, and visualization. C.B.: supervision, project administration. T.H.: conceptualization, methodology, resources, data curation, formal analysis, visualization, supervision, validation, writing—review and editing, and project administration. All authors have read and agreed to the published version of the manuscript.

**Funding:** This work was supported by grants from the Frontier Research Institute of Materials Science (FRIMS) of the Nagoya Institute of Technology and the Deutsche Forschungsgemeinschaft under GRK2495/E.

**Institutional Review Board Statement:** Not applicable.

**Informed Consent Statement:** Not applicable.

**Data Availability Statement:** The datasets used and/or analysed during the current study are available from the corresponding author upon reasonable request.

**Conflicts of Interest:** There are no conflict to declare.

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
