# Peer review of "X-ray Diffraction, Micro-Raman and X-ray Photoemission Spectroscopic Investigations for Hydrothermally Obtained Hybrid Compounds of Delafossite CuGaO2 and Wurtzite ZnO"

_ceramics, doi:10.3390/ceramics5040048_

Round 1

Reviewer 1 Report

The article by Minuk Choi et al. presents a study of crystalline heterostructures of CuGaO2 and ZnO obtained by hydrothermal deposition of ZnO on hexagonal platelet crystals of CuGaO2. The relevance of the research is in the search for semiconductor materials that can be used in solar cells, to increase the effectiveness of solar energy generation. The manuscript is relevant to this field and presented in a well-structured form. The manuscript is scientifically sound and the amount of research carried out is sufficient to substantiate the conclusions made. In general, the material will be of interest to the Coating's readers. Some comments on improving the article: it is worth including Table S1 in the main text of the article. In conclusion, indicate whether the objectives of the work have been achieved, what significance the results have for optoelectronic applications.

Author Response

Thank you for reviewing our manuscript and giving your positive comments. We revised our manuscript along with the comments, as follows.

(1)“it is worth including Table S1 in the main text of the article.”

Table S1 was moved to the maintext and labeled as Table 1. And correspondingly old Tables 1-4 were renumbered as Tables 2-5.

(2)“In conclusion, indicate whether the objectives of the work have been achieved, what significance the results have for optoelectronic applications.”

At the end of Conclusion of the revised manuscript, the following sentence was added to fulfill the reviewer’s criteria.

“..modified Auger parameters, indicating effective transport of electron carriers to the ZnO region from the hetero-interface. It was concluded that the hydrothermal approach for the hybrid compound synthesis was promising and could cut out a new path for the development of optoelectronic devices.”

All of the revisions are marked in red ink in the revised manuscript.

Reviewer 2 Report

The manuscript "X-ray diffraction, micro-Raman and X-ray photoemission spectroscopic investigations for hydrothermally obtained hybrid compounds of delafossite CuGaO2 and wurtzite ZnO" fulfills all the requirements.

I recommend to be accepted as it is

Author Response

Thank you for reviewing the manuscript and assessing our research work positively. We are happy to hear that for our manuscript submitted to Ceramics.
